# Characteristics of Technical and Tactical Preparation of Elite Judokas during the World Championships and Olympic Games

**DOI:** 10.3390/ijerph18115841

**Published:** 2021-05-29

**Authors:** Wiesław Błach, Łukasz Rydzik, Łukasz Błach, Wojciech J. Cynarski, Maciej Kostrzewa, Tadeusz Ambroży

**Affiliations:** 1Faculty of Physical Education & Sport, University School of Physical Education, 51-612 Wroclaw, Poland; wieslaw.judo@wp.pl (W.B.); blachlukas@gmail.com (Ł.B.); 2Institute of Sports Sciences, University of Physical Education, 31-571 Krakow, Poland; 3Institute of Physical Culture Studies, College of Medical Sciences, University of Rzeszow, 35-959 Rzeszów, Poland; ela_cyn@wp.pl; 4Department of Sports Training, Jerzy Kukuczka Academy of Physical Education in Katowice, 40-065 Katowice, Poland; m.kostrzewa@awf.katowice.pl

**Keywords:** judo, martial arts, technical and tactical preparation, fighting rules

## Abstract

The basis for achieving success in sport is technical preparation supported by adequate level of physical fitness. During judo competitions, athletes use technique to meet tactical objectives aimed to achieve victory. The modification of the rules of combat in judo that has been carried out in recent years has changed the course of competition. It seems to be interesting if there are relations between technical and tactical preparation expressed by means of indices and modification of the course of the fight caused by changes in the rules. The purpose of the paper was to determine the values of technical and tactical preparation of judokas during competition at the elite level. A hundred and twenty bouts during the Olympic Games in London in 2012 as well as 136 bouts fought during the World Championships in Rio de Janeiro in 2013 were analyzed. Verification was performed by calculating indicators of technical and tactical preparation. The results show a significant correlation between the indicators of technical and tactical preparation and the ranking in the general classification of the analyzed competitions. There were no statistically significant correlations between the change of fighting rules and the level of the examined indices of technical and tactical preparation. The results of the study verified the appropriate method of preparation for the competitions analyzed.

## 1. Introduction

Judo is a combat sport that includes many techniques of the fight and requires proper motor and physiological preparation [1]. This sport features open sensory and motor abilities with a dynamic sequence of events, and success is a result of the efficiency of the athlete’s actions and the number of the rival’s errors [2]. The analysis of the different techniques of performing, as well as many more details related to somatic and anthropometric changes during the athlete’s development, makes sports training more and more sophisticated, while the coaching staff are supported by specialists from various fields of science, and the athlete becomes a source of valuable feedback [3,4,5]. Moreover, judo bouts force versatile technical and tactical preparation on competitors, which is the core of sports and decides sports championships [6,7].

According to the experts in the field, it is not possible to achieve excellent results in combat sports without earlier technical preparation [8,9,10]. To rationally plan the training process, it is necessary to recognize which techniques and tactical activities are effective in modern sport judo fights [11]. The most significant factors contributing to success in sports include proper technical and tactical preparation as well as the ability to apply them efficiently [12].

Currently, there are many up-to-date developments that analyze judo bouts in terms of technical and tactical actions [10,13,14,15,16]. Gutiérrez-Santiago et al. [7] analyzed the fights during the World Championships in judo in 2017 in the 66–73 kg weight division. The indicators of technical and tactical training in terms of activity, efficiency, and effectiveness of attack operations of judokas of national teams of Japan and Russia fighting during the World Championships in 2013–2015 were also analyzed [9]. Maduro et al. [17] estimated the technical profile of judokas and their coaches. The relationship between the short-term changes of results and the results achieved in World Championships was also subject to study [18]. The analysis of bouts fought during the Olympic Games found a relationship between the increase in the values of technical–tactical indicators and success in elite judo [19]. Calmet et al. [20] also showed the evolution of judo caused by the impact of new competition rules. As the result of the study, the variables of technical and tactical training of the participants (the weight divisions all together) were determined.

In the examination of the variables affecting the efficiency of the fighting technique, and consequently the level of sporting achievements, attempts have been made to verify the athlete’s technical parameters in terms of activeness during competitions [21,22]. The course of the fight is determined by the rules affecting the quantitative and qualitative structure. When describing the sports fight in judo, one should mention such elements as the actual time of the fight, the time of work and rest, the number of technical and tactical activities and their types, activity, efficiency, and effectiveness [23,24,25]. The analysis of judo fighting conducted to date indicates the necessity to divide it into offensive and defensive fighting systems [26,27]. Both actions in attack and effective defense affect the final result of the fight, providing the athlete with points [28,29]. In planning a rational training program to ensure the achievement of the sports champion level, one should use the information obtained during the fight monitoring [30]. The most common form of determining the sports skill level of an athlete is the values of indices of technical and tactical preparation, which are used for training correction and determining the current skill level. The research on technical and tactical indicators has been the subject of many studies on combat analysis in Judo [31,32,33]. Monitoring of the changes within the sports fight in judo following the modification of the sports’ rules seems to be an important part of the research [24].

The research was inspired by the change of the rules introduced before the World Championships in Rio. The main changes included:(a)techniques using catching a competitor’s lower limbs below the belt were banned (in 2012, they were allowed if they were used after a different allowed technique);(b)the time limit of a golden score (overtime) was cancelled;(c)durations of holds were shortened;(d)many limitations in fighting for a hold were introduced.

They were supposed to make the fight more spectacular, attract media, and bring it closer to the original traditional version. Calculation and compilation of the values of indicators of technical and tactical preparation of judokas during competition at the elite level allows for the determination of the current level of competition in terms of the preparation of competitors and changes in the course of judo fights following the introduction of new regulations, and also for linking it with technical and tactical skill level [34].

The purpose of this paper was to determine the indicators of the technical and tactical skill level of judokas during competition at the elite level in two competitions of the highest rank, in between which a modification of the rules was made.

Answers for the following study questions were sought:(1)What are the values of the indicators of technical and tactical preparations of judokas competing in the World Championships in Rio de Janeiro in 2013?(2)What are the values of the indicators of technical and tactical preparations of judokas competing in the Olympic Games in London in 2012?(3)What are the differences in the indicators between the members of both study groups?

The research hypothesis assumes that modifications in judo fighting rules may have led to changes in the level of technical and tactical training indicators of elite judo athletes.

## 2. Materials and Methods

The study was based on the analyses of bouts fought during the Olympic Games in London in 2012 and the World Championships in Rio de Janeiro in 2013. The participants of the study were medalists in seven weight divisions. Technical and tactical actions taken by 56 judokas included 28 athletes participating in the Olympic Games and 28 judokas that took part in the World Championships. Altogether, 256 men’s fights were registered, including 120 bouts fought during the Olympics and 136 bouts fought during the World Championships. To determine the scores and places in the general classification, in addition to video analysis, we reviewed referee documentation and tournament reports made available by the International Judo Federation.

Age and height of the participants are shown in Table 1. Body height and age were determined based on anthropometric data from competition entry forms obtained from the International Judo Federation (IJF).

The analysis included judokas of different weight divisions who had various numbers of bouts fought during the competition (Table 2).

Sports fight analysis was performed by three champion-level judo coaches, based on the digital recording of selected tournament fights of the athletes studied. The recording was made with three cameras (Sony HDR-CX115, Manufacturer, Tokyo, Japan). Movavi Video Editor 14 software (Movavi, Wildwood, MO, USA) was used to merge the images. The setting of cameras allowed continuous observation of the athletes, referees, and the scoreboard. A single sheet was developed as the essential research tool. The sheet consisted of a time line of the fight, on which all the performed techniques were marked with symbols [35] at the appropriate time points, along with the number of points scored for a given technique. Data from the worksheets were entered into Excel (Microsoft, Redmond, WA, USA). Then, the values of technical and tactical preparation indices were calculated.

Variables of the technical and tactical training were determined based on videos of the bouts, and the computations were made of the following, Equations (1)–(6) [10]:The efficiency of the attack
(1)$$Sa=\frac{\left(M\times 5\right)+\left(M\times 7\right)+\left(\;M\times 10\;\right)\;}{N}$$

M—number of attacks awarded according to the attack performed (yuko—5, wazari—7, ippon—10)

*Sa*—the efficiency of the attack

*N*—number of observed bouts

2.The efficiency of the defense

So  is equal to the value of the indicator of the efficiency of the attack of the opponent of a given fighter.

3.Total efficiency

(2)Sk=Sa−So
where Sa  is the indicator of the efficiency of the attack and  So  is the indicator of the efficiency of the defense.

*Sk*—total efficiency

4.The effectiveness of the attack

(3)$$Ea=\frac{\mathrm{number}\;\mathrm{of}\;\mathrm{effective}\;\mathrm{attacks}}{\mathrm{number}\;\mathrm{of}\;\mathrm{all}\;\mathrm{attacks}}\times 100$$

*Ea*—Effective attack is an offensive activity that scored points. An attack is any offensive action.

Number of all attacks is the sum of all athlete’s actions that are offensive in nature

5.The effectiveness of the defense

(4)$$Eo=\left(\;1-\frac{As}{Ap}\right)\times 100$$

*Eo*—the effectiveness of the defense

Where *As* is a number of effective attacks performed by the opponent and *Ap* is the number of all attacks by the opponent.

6.The activeness of the attack

(5)$$Aa=\frac{\mathrm{number}\;\mathrm{of}\;\mathrm{recorded}\;\mathrm{attacks}\;\mathrm{of}\;\mathrm{the}\;\mathrm{competitor}}{\mathrm{number}\;\mathrm{of}\;\mathrm{bouts}}$$

7.The activeness of the defense

Ao  is equal to the value of the indicator Aa of the activeness of the attack of the opponent.

8.Total activeness

(6)A=Aa – Ao

*A*—total activeness

*Aa*—the activeness of the attack

*Ao*—the activeness of the defense

### Methods of Statistical Analysis

The statistical analysis of the data was made using R ver. 3.6.3. The chosen variables were the arithmetic mean with standard deviation (± SD).

The differences between paired data (London and Rio comparison) were analyzed with the paired Students t-test or Wilcoxon test. Before the appropriate method was selected, the assumption about normality of distribution was verified for the difference between both features with the Shapiro–Francia test. When the test confirmed the normality of the distribution, the Students t-test was selected for analyzing the difference. Otherwise, the Wilcoxon test was used.

Firstly, it was verified whether both variables were normally distributed. If non-normality was detected, the Spearman correlation test was applied. Otherwise, Pearson’s test was used. For all tests, we assumed the significance of *p* < 0.05.

In this paper, a single judoka is a statistical unit in both paired (comparison between London and Rio) and non-paired tests. Each competitor’s statistics are computed based on all his bouts.

## 3. Results

Most of the indicators turned out to be higher during the World Championships (Rio de Janeiro 2013, Brazil). The only lower value compared to the Olympic Games (London 2012, UK) was recorded for the efficiency. There were no statistically significant differences in all indicators measured (*p* > 0.05) (Table 3).

There are several significant relations between the described variables. Total activeness is moderately positively related to the activeness of the attack and negatively related to the activeness of the defense (0.69 and −0.54, respectively). Total activeness and the activeness of the attack are negatively correlated to the effectiveness of the attack (−0.36 and −0.37, respectively). Activeness of the attack is also positively correlated to the efficiency of the attack (0.33). The efficiency of the defense is negatively correlated to both effectiveness of the defense and total efficiency (−0.82 and −0.33, respectively) (Table 4).

There is a positive relationship between the rank of a competitor and the values of the efficiency of the defense (So) for London (0.45) and Rio (0.5) competitions as well as for the competitions altogether (0.47). There is a moderate negative correlation between total efficiency (S) and the rank in Rio (−0.54) and both competitions together (−0.41). There is a similar relationship between the effectiveness of the defense (Eo) and the rank in Rio (−0.42) and both contests (−0.31) (Table 5).

## 4. Discussion

This paper aimed to evaluate the indicators of the level of technical and tactical skills in elite judokas in competitive settings, i.e., the World Championship and Olympic Games, with the fighting rule change occurring between the two events.

The indicator of the efficiency of the attack in both groups was different, but the differences were not statistically significant. Therefore, it should be concluded that the modification of the rules did not affect the efficiency of the athlete’s attack. Such results also indicate that changes in regulations do not significantly influence the evolution of fighting techniques in Judo. Other findings are presented in the analysis of rule change and performance evolution in judo based on the analysis of Olympic Games in 2012 and 2016 and World Championships in 2015 and 2017 at which there were statistically significant differences in terms of technical and tactical preparation indicators [20]. Similarly, Adam et al. analyzed the efficiency of offensive technical actions of athletes in the context of changes in the sports fighting rules. These authors analyzed championship tournaments from 2008–2012. Based on the observations in 2008, 2009, and 2012, they presented the technical evolution of judo in terms of the most efficient technical actions. Both the development of the sport and the changes in the fighting rules resulted in shifting the burden of combat towards hand techniques (throws from the te-waza group). The values of efficiency indicators during this period were almost twice smaller than those presented in this paper and were 2.88, 3.46, and 2.35, respectively [10]. Such a difference indicates an increase in the technical and tactical skill levels of judo players. The efficiency of the defense during the Olympic Games was 1.24, while during the World Championships, it reached a higher value (1.44). There is a noticeable difference, but it is not statistically significant. Therefore, it can be considered that the elimination of lower limb catching does not significantly affect the course of the fight. No statistically significant differences were also observed in the remaining indicators of technical and tactical preparation (activeness, effectiveness of the attack, and defence). The course of the fight does not depend on the changes in the rules, and it can be assumed that it is determined by the level of technical skills and the method of implementation of the training process.

A statistically significant positive correlation was found between the medal positions and the efficiency of defense during the world championships, Olympic Games, and without the division into groups. This demonstrates the crucial role of defense, which appears to be a necessary factor for success [23]. Defensive actions are a fundamental factor in athletic competition in grappling sports [36]. The efficiency of the defense is understood to mean defending the opponent’s attack even with his or her high efficiency of the attack. Not allowing the opponent to perform offensive actions is the basis for winning in combat sports. In judo, a large part of the preparatory period during the pre-competition mesocycle is devoted to the technical and tactical training of defensive actions [37].

In the present study, there was a negative relationship between total efficiency and the efficiency of the defense and medal position in the Rio Championships and without division into groups. This demonstrates that a higher index of technical and tactical preparation corresponds to a lower medal position, i.e., to a better final result. Therefore, the accumulation of the efficiency of the attack, efficiency of the defense, and the effectiveness of the defense is an essential factor in the success of an athlete. According to Sterkowicz [38], top-class judokas should have a wide range of tactical actions, and in particular, be able to force penalties on the opponent. Franchini’s research [39] indicated that top-class judokas use significantly more technical actions in different directions that lead to a scoring advantage. Such results confirm the conclusions of our research. They show that the number of different technical actions and the variety of sides on which they were performed depended on the number of fights won and ippon scored. Therefore, the greater number of technical actions used and the variation in the directions of attack seem to be important factors causing unpredictability during judo fights.

In the analysis of the relationships between indicators of technical and tactical preparation of judo competitors in elite championships (World Championships and Olympic Games), the influence of changes in the rule change in terms of technique and the limitation of the number of allowed actions was mainly analyzed, without finding any relations between these variables. Smaruj et al. [40] analyzed 1643 fights of female judo players that took place between 2001 and 2004 during national classification tournaments. These authors emphasized that the effective duration of a women’s judo bout has changed in 2002 from 5 min to 4 min [40]. This change could have affected the values of the indicators of technical and tactical preparation achieved by the athletes. It can be presumed that changes in technique were compensated for by the players with a high level of physical fitness and by replacing forbidden techniques with others. Therefore, preparation indicators did not change, whereas shortening of the fight time caused the necessity of changing the physiological profile of the female players. The necessity to modify the method of endurance training caused by this fact could have influenced the profile of technical and tactical preparation [15].

The high positive correlation between the effectiveness and efficiency of the attack and the negative correlation with activeness may indicate a trend caused by the evolution of judo’s way of fighting. New combat sport rules force fighters to strive for high point advantage [41]. This can result in forcing athletes’ activity using penalties and limiting defensive options as a result of the ban on blocking by catching the opponent judogi’s leg or lower limbs with upper limbs below the waist [42]. It should be added that the effectiveness of the attack (Ea) and the effectiveness of the defense (Eo) are characterized by a high universality, consisting in an objective characterization of the technical actions of competitors, regardless of the number of analyzed fights and the duration and the way the fight is resolved [10].

Therefore, to optimize the coaching process in combat sports, the emphasis should be on the development of natural aptitudes and methods of fighting, depending on functional potential. During the bout, motor abilities are most often manifested in a complex manner. Therefore, the development of technical and tactical skills in individual athletes should take into consideration their profile of motor abilities.

### Limitation of the Study

Nonetheless, the findings of this study have to be seen in light of some limitations. The indicators determining the level of technical and tactical preparation of judokas were analyzed only during two competitions of different ranks. Thus, it may be speculated that if competitions of the same rank were compared (Olympic Games vs. Olympic Games), different results would be obtained. Additionally, there was about a year of differences between the compared occupations; it is possible that a comparison of competition with a longer time interval would show the impact of training or tactical preparation to a greater extent. Moreover, only elite male judokas were analyzed, and it cannot be excluded that these results may differ among females or sub-elite judokas.

## 5. Conclusions

Between the 2012 Olympic Games in London and the 2013 World Championships, there were no statistically significant differences in the level of technical and tactical preparation despite the change of fighting rules that took place between the competitions.

The level of technical and tactical preparation is closely related to the medal place achieved by the athletes at the elite level.

For achieving high medal positions in championships, more attention should be paid to the level of defensive actions in the process of sports training of judokas.

### Application

Because the changes in the rules of judo fights in terms of technique do not significantly differentiate the level of technical and tactical preparation of the competitors, modifications to the training process should go in the direction of the most comprehensive training of the player in terms of technique and physical fitness.

## Figures and Tables

**Table 1 ijerph-18-05841-t001:** Descriptive characteristics of the study participants.

Stats Features	Mean	± SD	Q1	ME	Q2	Min–Max
**Age (years)**	26.32	3.59	24	26	29	20–34
**Height (cm)**	178.42	11.48	170.00	176	188	160–204

SD—standard deviation, Q1—first quartile, ME—median, Q2—second quartile, Min–Max—minimum–maximum.

**Table 2 ijerph-18-05841-t002:** Number and duration of bouts fought during the Olympic Games in London in 2012 and the World Championships in Rio de Janeiro in 2013, including weight divisions.

Weight Division	Olympic Games London	World Championships Rio
	Number of Bouts	Number of Bouts
60 kg	16	20
66 kg	16	21
73 kg	18	22
81 kg	18	21
90 kg	18	18
100 kg	17	18
+100 kg	17	16
Σ	120	136

**Table 3 ijerph-18-05841-t003:** The efficiency, activeness, and effectiveness of the attack and defense in London and Rio.

	Indicator	Mean	SD	Q1	ME	Q2	Min–Max	*p*
The efficiency of the attack (Sa)	Sa London	5.92	2.49	4.00	5.20	8.10	2.00–10.02	0.083
	Sa Rio	7.79	2.47	5.90	7.91	9.75	3.50–13.83	
The efficiency of the defense (So)	So London	1.24	1.04	0.00	1.08	2.00	0.00–3.40	0.715
	So Rio	1.44	1.29	0.00	1.53	2.07	0.00–4.50	
Total efficiency (Sk)	Sk London	4.68	2.57	2.96	4.45	6.50	−0.60–10.00	0.233
	Sk Rio	6.35	2.77	4.41	6.36	7.83	1.50–13.83	
The activeness of the attack (Aa)	Aa London	1.52	0.61	1.01	1.61	1.93	0.47–3.00	0.063
	Aa Rio	1.81	0.54	1.50	1.80	2.10	0.77–3.23	
The activeness of the defense (Ao)	Ao London	1.23	0.51	0.93	1.21	1.59	0.30–2.30	0.162
	Ao Rio	1.60	1.52	1.29	1.52	1.70	1.07–2.89	
Total activeness (A)	A London	0.29	0.59	0.01	0.23	0.65	−0.74–1.61	0.73
	A Rio	0.21	0.77	-0.45	0.11	0.71	−1.21–1.73	
The effectiveness of the attack (Ea)	Ea London	12.96	6.51	7.78	13.33	17.26	3.33–26.31	0.269
	Rio	17.25	7.81	11.58	16.32	24.16	5.76–33.33	
The effectiveness of the defense (Eo)	Eo London	76.78	2.70	95.11	96.87	100.00	90.90–100.00	0.927
	Rio	96.40	3.47	94.42	96.42	100.00	85.00–100.00	

SD—standard deviation, Q1—first quartile, ME—median, Q2—second quartile, Min-Max—minimum–maximum.

**Table 4 ijerph-18-05841-t004:** Correlation coefficient for indicators of technical and tactical preparations.

Variable	Correlation Coefficient, No Weight Divisions,							
	Sa	So	Sk	Ea	Eo	Aa	Ao	A
Sa	-	0.09	0.91	0.61	**−0.05**	0.33	0.31	0.06
So	0.09	-	−0.33	−0.18	−0.82	0.21	0.44	−0.14
Sk	0.91	−0.33	-	0.65	0.28	0.23	0.11	0.12
Ea	0.61	−0.18	0.65	-	0.09	−0.37	**0.04**	−0.36
Eo	**−0.05**	−0.82	0.28	0.09	-	−0.04	**−0.03**	**−0.05**
Aa	0.33	0.21	0.23	−0.37	**−0.04**	-	0.23	0.69
Ao	0.31	0.44	0.11	**0.04**	**−0.03**	0.23	-	−0.54
A	0.06	−0.14	0.12	−0.36	**−0.05**	0.69	−0.54	-

Sa—the efficiency of the attack, So—the efficiency of the defense, Sk—total efficiency, Ea—the effectiveness of the attack, Eo—the effectiveness of the defense, Aa—the activeness of the attack, Ao—the activeness of the defense, A—the activeness; Statistically significant values are in bold (*p* < 0.05).

**Table 5 ijerph-18-05841-t005:** Relationship between the indicators of technical and tactical preparations and the rank in the general classification of the analyzed competitions.

Spearman’s Rank Correlation Coefficient	Sa	So	S	Ea	Eo	Aa	Ao	A
OG 2012 London	−0.23	**0.45**	−0.33	−0.11	−0.20	−0.06	0.24	−0.30
WC 2013 Rio	−0.29	**0.50**	**−0.54**	−0.33	**−0.42**	0.08	0.20	−0.10
Altogether	−0.23	**0.47**	**−0.41**	−0.20	**−0.31**	0.00	0.22	−0.18

Sa—the efficiency of the attack, So—the efficiency of the defense, S—total efficiency, Ea—the effectiveness of the attack, Eo—the effectiveness of the defense, Aa—the activeness of the attack, Ao—the activeness of the defense, A—the activeness; Statistically significant values are in bold, Coefficients in bold are significant *p* < 0.05, OG—Olimpic Game, WC—World Championship.

## Data Availability

The data presented in this study are available on request from the corresponding author.

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
