# Peer review of "Characteristics of Technical and Tactical Preparation of Elite Judokas during the World Championships and Olympic Games"

_ijerph, 2021, doi:10.3390/ijerph18115841_

Round 1

Reviewer 1 Report

Significant variables determining the level of technical and tactical preparation of judokas for competitions at the elite level

Manuscript Review

Abstract:

It would be beneficial to include some basic participant characteristics in the abstract.

Additionally, it would be beneficial to add some specific results within the abstract.

Overall, the abstract is quite vague and does not give a great overview of the entire manuscript.

Introduction:

Page 1, line 32-33: Revise English language “many techniques of the fight, requiring proper motor…”

Page 1, line 35: Revise the English language as the sentence feels a little disjointed.

Page 1, line 37-38: Change the to an – “anthropometric changes during an athlete’s development…”

Page 1, line 43-44: Suggest to either add more citations or rephrase the statement – I’m not quite sure how a statement such as “experts in the field” is only supported by one citation.

Page 2, line 49: Correct the typo – efficientlyso.

Page 2, line 50: Support the statement with citations – You can’t state “there are many up-to-date developments in terms of technical and tactical actions.” without the supporting citations.

Page 2, line 63: Revise English language “As a result of these studies…”

Page 2, line 65: Remove ‘study’ – “Answers for the following questions were sought:”

Page 2, lines 74-80: You mention the rule changes in Judo and how the research was inspired by this, it would be beneficial to discuss in minor detail the reasoning behind the rule changes. This will help lay a better foundation for the paper and gives better meaning to the study purpose.

Overall, the introduction lacks impact. The authors need to dig a little deeper into some of the research they have introduced to better set the scene for the purpose of their manuscript. Simply stating ‘x did this, and y did that’ does not lay a sufficient foundation for the manuscript.

Materials and Methods:

Page 2, lines 90-91: The purpose should be clearly stated in the introduction, before the hypotheses. Please revise accordingly.

Page 2: There is no mention of informed consent in the methods (or the entire manuscript). Please revise to state informed consent was obtained OR clearly state why informed consent was not required.

Page 3, Table 1: Please replace the mean symbol for the word ‘mean’ as the symbol is unclear, OR revise the symbol to make it clearer.

Page 3, Table 1: Please define Q1, ME, and Q2.

Page 4, line 142: Please clarify which tables – “The results are presented and compared in Tables # - #).”

Page 4, lines 146 & 150: Please revise to state ‘Students t-test’ not t-Students test.

Maybe a personal preference, but I would mention the assumption tests first, and then describe the appropriate statistical analyses following the assumption checks.

Results:

I would revise and simplify the results section as it is a little clunky and wordy.

i.e. Page 4, lines 161-165: “There was no difference (p>0.05) in the efficiency of the attack (Sa) between medalists at the Olympic Games in London 2012 (5.92 ± 2.49) and medalists at the World Championships in Rio (7.79 ± 2.47).”

Page 5, Table 3: Please replace the mean symbol for the word ‘mean’ as the symbol is unclear, OR revise the symbol to make it clearer.

Page 5, Table 3: Revise to report ME (not Me).

Just a point of clarity – is there any need for Table 3 if all of these results are reported in text and they are all non-significant? Either simplify the written results, or remove Table 3.

Page 6, lines 223-225: Please remove this statement from the results section as it is not appropriate to discuss results in this section – simply state the observed relationships. This sentence would better fit within the discussion section of the manuscript.

Page 6, Table 4 and Table 5: Please be consistent in how you report significance. In Table 4, significance is in red, however, in Table 5, significance is bolded.  Revise one or the other to be consistent in formatting.

Discussion:

Please start by restating the purpose and mention the key findings of the study.

Page 6, lines 229-230: Please remove this statement as you can’t say it was higher if it was not supported by statistical significance.

Please revise the entire opening paragraph of the Discussion – You can’t suggest that observed differences may have been a result of the rule changes if there are NO significant differences observed.

Page 6, lines 240-241: Were the differences observed by Adams et al. statistically significant? Please report this, otherwise remove.

Page 7, lines 250-258: Are the differences observed in the Smaruj study statistically significant?

Page 7, lines 260-261: Please rephrase as it is incorrect to state an “insignificant increase”. If the difference is not statistically significant, then essentially those reported values are statistically the same.

Page 7, lines 270-278: I would spend more time discussing these observed relationships, rather than simply restating them. What do they mean? How can coaches/athletes utilize this information? Has other research made similar observations? At present, all you do is restate the observed correlations, it would be beneficial to the manuscript to discuss these in more detail.

Page 7, line 294: Please add a space between ‘was about’.

Conclusion:

The entire conclusion needs to be revised to better reflect the lack of statistical significance in the findings.

The authors can’t make statements such as “Competitors fighting at the world championships …. showed higher levels of …” if it is not supported by the appropriate statistical significance.

Citations:

Please double-check your reference list for consistency in formatting (Reference 9 is ALL CAPITAL).

Author Response

Dear Reviewer,

Thank you very much for your time and valuable comments, which all have been considered and incorporated. The detailed list of responses is given below. We hope that the modifications and explanation will be acceptable for you.

Yours sincerely,

Rydzik, corresponding author

Abstract:

It would be beneficial to include some basic participant characteristics in the abstract.

Additionally, it would be beneficial to add some specific results within the abstract.

Overall, the abstract is quite vague and does not give a great overview of the entire manuscript.

A: We have revised the abstract to more closely reflect the content of the manuscript. We have also added basic characteristics of study participants (sports skill level) and figures.

Page 1, line 32-33: Revise English language “many techniques of the fight, requiring proper motor…”

A: This has been defined

Page 1, line 35: Revise the English language as the sentence feels a little disjointed.

A: This has been defined

Page 1, line 37-38: Change the to an – “anthropometric changes during an athlete’s development…

A: This has been defined

Page 1, line 37-38: Change the to an – “anthropometric changes during an athlete’s development…”

A: This has been defined

Page 1, line 43-44: Suggest to either add more citations or rephrase the statement – I’m not quite sure how a statement such as “experts in the field” is only supported by one citation.

A: More references have been added

Page 2, line 49: Correct the typo – efficientlyso.

Page 2, line 50: Support the statement with citations – You can’t state “there are many up-to-date developments in terms of technical and tactical actions.” without the supporting citations.

A: Dodano

Page 2, line 63: Revise English language “As a result of these studies…”

A: This has been defined

Page 2, line 65: Remove ‘study’ – “Answers for the following questions were sought:”

A: This has been defined

Page 2, lines 74-80: You mention the rule changes in Judo and how the research was inspired by this, it would be beneficial to discuss in minor detail the reasoning behind the rule changes. This will help lay a better foundation for the paper and gives better meaning to the study purpose.

A: The rule changes have been described. The rule changes were introduced as an initiative of the International Judo Federation. They were supposed to make the fight more spectacular, attract media, and bring it closer to the original traditional version proposed by the kodokan, with exclusion of the catching of the competitor’s limbs.

Overall, the introduction lacks impact. The authors need to dig a little deeper into some of the research they have introduced to better set the scene for the purpose of their manuscript. Simply stating ‘x did this, and y did that’ does not lay a sufficient foundation for the manuscript.

A: The introduction section has been revised

Page 2, lines 90-91: The purpose should be clearly stated in the introduction, before the hypotheses. Please revise accordingly.

A: This part has been changed following the Reviewer's suggestions

Page 2: There is no mention of informed consent in the methods (or the entire manuscript). Please revise to state informed consent was obtained OR clearly state why informed consent was not required.

A: The research concerned the analysis of a sports fight and was based on multimedia recording of fights and, therefore, informed consent was not required. Furthermore, they did not involve action on a specific player

 Page 3, Table 1: Please replace the mean symbol for the word ‘mean’ as the symbol is unclear, OR revise the symbol to make it clearer.

A: This has been corrected

Page 3, Table 1: Please define Q1, ME, and Q2.

A: This has been defined

Page 4, line 142: Please clarify which tables – “The results are presented and compared in Tables # - #).”

A: This has been added 

Page 4, lines 146 & 150: Please revise to state ‘Students t-test’ not t-Students test.

A: This has been corrected

Maybe a personal preference, but I would mention the assumption tests first, and then describe the appropriate statistical analyses following the assumption checks.

A: This has been corrected

i.e. Page 4, lines 161-165: “There was no difference (p>0.05) in the efficiency of the attack (Sa) between medalists at the Olympic Games in London 2012 (5.92 ± 2.49) and medalists at the World Championships in Rio (7.79 ± 2.47).”

A: This part has been corrected

Page 5, Table 3: Please replace the mean symbol for the word ‘mean’ as the symbol is unclear, OR revise the symbol to make it clearer.

A: This part has been corrected

Page 5, Table 3: Revise to report ME (not Me).

A: This part has been corrected

Just a point of clarity – is there any need for Table 3 if all of these results are reported in text and they are all non-significant? Either simplify the written results, or remove Table 3.

A: We propose to leave Table 3 due to the request of other Reviewers. Furthermore, the text has been simplified as suggested

Page 6, lines 223-225: Please remove this statement from the results section as it is not appropriate to discuss results in this section – simply state the observed relationships. This sentence would better fit within the discussion section of the manuscript.

A: This part has been removed

Page 6, Table 4 and Table 5: Please be consistent in how you report significance. In Table 4, significance is in red, however, in Table 5, significance is bolded.  Revise one or the other to be consistent in formatting.

A: the formatting has been corrected

Please start by restating the purpose and mention the key findings of the study.

A: This part has been corrected

Page 6, lines 229-230: Please remove this statement as you can’t say it was higher if it was not supported by statistical significance.

A: This part has been corrected

Discussion:

Please start by restating the purpose and mention the key findings of the study.

A: The discussion section has been rewritten

Page 6, lines 229-230: Please remove this statement as you can’t say it was higher if it was not supported by statistical significance.

A: The discussion section has been rewritten

Please revise the entire opening paragraph of the Discussion – You can’t suggest that observed differences may have been a result of the rule changes if there are NO significant differences observed.

A: The discussion section has been rewritten

Page 6, lines 240-241: Were the differences observed by Adams et al. statistically significant? Please report this, otherwise remove.

A: The discussion section has been rewritten

Page 7, lines 250-258: Are the differences observed in the Smaruj study statistically significant?

A: The discussion section has been rewritten

Page 7, lines 260-261: Please rephrase as it is incorrect to state an “insignificant increase”. If the difference is not statistically significant, then essentially those reported values are statistically the same.

A: The discussion section has been rewritten

Page 7, lines 270-278: I would spend more time discussing these observed relationships, rather than simply restating them. What do they mean? How can coaches/athletes utilize this information? Has other research made similar observations? At present, all you do is restate the observed correlations, it would be beneficial to the manuscript to discuss these in more detail.

A: The discussion section has been rewritten

Page 7, line 294: Please add a space between ‘was about’.

A: The discussion section has been rewritten

Conclusion:

The entire conclusion needs to be revised to better reflect the lack of statistical significance in the findings.

The authors can’t make statements such as “Competitors fighting at the world championships …. showed higher levels of …” if it is not supported by the appropriate statistical significance.

A:  New conclusions have been drawn

Citations:

Please double-check your reference list for consistency in formatting (Reference 9 is ALL CAPITAL).

A: This part has been corrected

Reviewer 2 Report

The manuscript entitled “Significant variables determining the level of technical and tactical preparation of judokas for competition at the elite level” presents an analysis of a large number of judo fights in elite judokas, specifically the tactical and technical variables. The whole manuscript is interesting and the writing has a good flow, but there are considerations the authors have to take into consideration:

ABSTRACT:

If the limit in the number of words allows it, it could be interesting to include in the methods section the variables analyzed and the procedures for analyzing the bouts. The information provided in the background might be less important for an abstract.

The writing of the conclusions section seems unclear and with some spelling mistakes, please review this part.

Keywords: authors should include keywords more adequate for this study, avoiding general terms such as “analysis” or “indices”.

INTRODUCTION:

Line 49, before the reference number 10, please solve this spelling mistake.

Line 81-88. All the introduction presents very good flow and relevant data to provide a background for the issue, but this last paragraph seems unclear to present the aim of the study.

MATERIALS AND METHODS:

Line 90, “the purpose of this paper was to determine the indicators determining…” authors might rewrite this sentence using other verbs such as define, state etc.

Table 1: It presents very relevant information. However, it could be interesting to include the meaning of the abbreviations in a footnote since the terms Q1, Q2 and ME have not been described in the text before. Moreover, in the second column the header is not readable, at least in the table in the reviewer’s version.

The authors describe correctly the variables of the study, but how they obtained that information? If the information was obtained by videos, seeing the bouts or any other way, it should be stated and explained in the methods section.

RESULTS: all the results section is clearly explained but the Table 3 repeats most of the information provided in text. The authors should review this entire section in order to include relevant information in text and in the table but without repeating the same information in both parts. Besides, provide the effects sizes of the pairwise comparations could increase the relevance of the results.

Again, the header of the second column in the Table 3 is unreadable, and authors should include the meaning of the abbreviations in a footnote below the table.

Table 4: the information provided in the second column should be provided in a footnote below the table (coefficients in red are significant with p < 0.05). In the footnote says the significant values are in bold but at least in the version provided to this reviewer there are no values in bold.

Table 5: the same as suggested for table 4, the information in header of the first column (coefficients in bold are significant with p < 0.05) should be below the table in a footnote. Moreover, it is necessary to include the abbreviations again, although they have been explained previously in the text or in other tables.

DISCUSSION:

The first paragraph has to be rewritten, since is incorrect to state that one variable is higher than other with no statistically significant differences. If the p value is > than 0.05 then there are no differences.

Until the line 259, the authors only describe the results found in this study, but they do not compare with prior investigations or provide explanations to them. This section should include more information besides the results of the study, yet provided in the results section. If the rules have changed, why there are no significant differences between groups? What are the differences between the variables? These differences could explain the unequal results?

CONCLUSIONS: Again, there are no statistical significant differences in the variables so it cannot be stated that competitors in 2013 showed higher levels than those in 2012. All the conclusions have to be rewritten according to this concept.

Line 311: this paragraph seems more suitable for a “practical applications” section but not for the conclusion.

Author Response

Dear Reviewer,

Thank you very much for your time and valuable comments, which all have been considered and incorporated. The detailed list of responses is given below. We hope that the modifications and explanation will be acceptable for you.

Yours sincerely,

Rydzik, corresponding author

ABSTRACT:

If the limit in the number of words allows it, it could be interesting to include in the methods section the variables analyzed and the procedures for analyzing the bouts. The information provided in the background might be less important for an abstract.

The writing of the conclusions section seems unclear and with some spelling mistakes, please review this part.

A: The abstract section has been corrected

Keywords: authors should include keywords more adequate for this study, avoiding general terms such as “analysis” or “indices”.

 A: Keywords have been corrected

Line 49, before the reference number 10, please solve this spelling mistake.

A: This has been corrected

Line 81-88. All the introduction presents very good flow and relevant data to provide a background for the issue, but this last paragraph seems unclear to present the aim of the study.

A : The aim of the study has been added

Line 90, “the purpose of this paper was to determine the indicators determining…” authors might rewrite this sentence using other verbs such as define, state etc.

A: This sentence has been moved to the Introduction and changed

Table 1: It presents very relevant information. However, it could be interesting to include the meaning of the abbreviations in a footnote since the terms Q1, Q2 and ME have not been described in the text before. Moreover, in the second column the header is not readable, at least in the table in the reviewer’s version.

A: This part has been corrected

The authors describe correctly the variables of the study, but how they obtained that information? If the information was obtained by videos, seeing the bouts or any other way, it should be stated and explained in the methods section.

A: The information has been added under Table 2

RESULTS: all the results section is clearly explained but the Table 3 repeats most of the information provided in text. The authors should review this entire section in order to include relevant information in text and in the table but without repeating the same information in both parts. Besides, provide the effects sizes of the pairwise comparations could increase the relevance of the results

A: The description of Table 3 has been rewritten and considerably shortened

Again, the header of the second column in the Table 3 is unreadable, and authors should include the meaning of the abbreviations in a footnote below the table.

A: This part has been corrected

Table 4: the information provided in the second column should be provided in a footnote below the table (coefficients in red are significant with p < 0.05). In the footnote says the significant values are in bold but at least in the version provided to this reviewer there are no values in bold.

A: This part has been corrected

Table 5: the same as suggested for table 4, the information in header of the first column (coefficients in bold are significant with p < 0.05) should be below the table in a footnote. Moreover, it is necessary to include the abbreviations again, although they have been explained previously in the text or in other tables.

A: This part has been corrected

DISCUSSION:

The first paragraph has to be rewritten, since is incorrect to state that one variable is higher than other with no statistically significant differences. If the p value is > than 0.05 then there are no differences.

Until the line 259, the authors only describe the results found in this study, but they do not compare with prior investigations or provide explanations to them. This section should include more information besides the results of the study, yet provided in the results section. If the rules have changed, why there are no significant differences between groups? What are the differences between the variables? These differences could explain the unequal results?

 A: The discussion has been rewritten as suggested by the Reviewer

CONCLUSIONS: Again, there are no statistical significant differences in the variables so it cannot be stated that competitors in 2013 showed higher levels than those in 2012. All the conclusions have to be rewritten according to this concept.

Line 311: this paragraph seems more suitable for a “practical applications” section but not for the conclusion.

A: The conclusions have been rewritten as suggested by the Reviewer.

Reviewer 3 Report

Dear Authors,

The study is interesting; however, several parts need to be addressed.

The title is unclear. Try to rewrite it.

Introduction:

From the introduction, It is not clear how you can measure the efficiency of attacks and defence in judo. There is no rationale why you chose this particular methodology in your methods. Add a paragraph. 

Your hypothesis at the end of the introduction is unclear and too general. The first sentence from the methods, Lines 90-93, reveals your aim and this sentence should be put at the end of the introduction. Correct

Methods:

The methods section is poorly written.

There are several important pieces of information missing that needs to be added:

-who analysed the videos (how many experts)?

-what software was used?

-what coding was used (who determined the coding - how many experts)?

-Where did you take the formulas for efficiency and the rest of your indicators for judo fight? References needed

-How and when was height measured? report

Table 1 - Q1, ME and Q2 abbreviations should be explained under the table

Table 2 - Why do you report the duration of fights? First - This data does not make any sense; Second - you don't use this data in your analysis. Delete

The formulas section is very unclear and hard to read. Restructure for greater clarity!

I the results - table 5 - you report rank in the general classification of the analyzed competitions. Where is this described in the methods? Where are the data reported?

Discussion:

Lines 246 - 249 are unclear and this is not a discussion. This is just stating jour findings. You are not explaining it nor connecting it to the previous studies, nor adding your own explanation. Rewrite!

The same repeated itself in Lines 270-275. Just a copy-paste from results. Discussion afterwards is not connected to your results.

Overall the paper is not well written, especially in the methods section.   Despite that, it represents interesting research and from my point of view it needs a major revision

Kind regards

Author Response

Dear Reviewer,

Thank you very much for your time and valuable comments, which all have been considered and incorporated. The detailed list of responses is given below. We hope that the modifications and explanation will be acceptable for you.

Yours sincerely,

Rydzik, corresponding author

The study is interesting; however, several parts need to be addressed.

The title is unclear. Try to rewrite it.

A: A new title has been proposed

Introduction:

From the introduction, It is not clear how you can measure the efficiency of attacks and defence in judo. There is no rationale why you chose this particular methodology in your methods. Add a paragraph. 

A: New paragraph has been added

Your hypothesis at the end of the introduction is unclear and too general. The first sentence from the methods, Lines 90-93, reveals your aim and this sentence should be put at the end of the introduction. Correct

A:  This part has been corrected

Methods:

The methods section is poorly written.

There are several important pieces of information missing that needs to be added:

-who analysed the videos (how many experts)?

A: This part has been explained

-what software was used?

A: Relevant information has been added

-what coding was used (who determined the coding - how many experts)?

A: Relevant information has been added

-Where did you take the formulas for efficiency and the rest of your indicators for judo fight? References needed

A: References have been added

-How and when was height measured? Report

A: Body height and age were determined based on anthropometric data from competition entry forms obtained from the International Judo Federation (IJF)

Table 1 - Q1, ME and Q2 abbreviations should be explained under the table

A: This part has been corrected

Table 2 - Why do you report the duration of fights? First - This data does not make any sense; Second - you don't use this data in your analysis. Delete

A: This has been deleted.

The formulas section is very unclear and hard to read. Restructure for greater clarity!

A: the results of formulas result from the information contained in references. The language has been improved.

I the results - table 5 - you report rank in the general classification of the analyzed competitions. Where is this described in the methods? Where are the data reported?

A: The method of data extraction has been complemented in the methods section

Discussion:

Lines 246 - 249 are unclear and this is not a discussion. This is just stating jour findings. You are not explaining it nor connecting it to the previous studies, nor adding your own explanation. Rewrite!

The same repeated itself in Lines 270-275. Just a copy-paste from results. Discussion afterwards is not connected to your results.

 A: The discussion section has been rewritten

Overall the paper is not well written, especially in the methods section.   Despite that, it represents interesting research and from my point of view it needs a major revision

A: A thorough revision of the discussion and introduction sections has been made.

Round 2

Reviewer 1 Report

The authors have put a lot of effort into addressing the point raised by the Reviewer, especially linked to English proofreading. Moreover, the discussion section appears to better reflect the results observed from the study. I feel that the manuscript has now improved.

Author Response

Thanks for your review

Reviewer 2 Report

This reviewer wants to congratulate authors for their changes in the manuscript, which largely increase the quality of the work presented.

Author Response

Thanks for your review

Reviewer 3 Report

Dear Authors,

Thank you for addressing the majority of the comments. However, some parts still need to be adequately addressed.

The abstract is poorly written. Rewrite and add the all-important findings of your research including that there were no statistically significant differences in the level of technical and tactical preparation despite the change of fighting rules that took place between the competitions.

Line 23-25: The respondents were at the highest world elite sports skill level and had had medal achievements at the World Championships and Olympic Games. / The sentence is too long and it does not clearly state what sample was used. Rewrite

Table 1 - Q1, ME, and Q2 abbreviations are in Polish? Translate
Table 2 - The head of the table is incorrect as you just state World Championships Rio. Correct and also add OG

You stated: ''Sports fight analysis was performed by three champion-level judo coaches'' So what were your criteria to be a champion-level judo coach? Why did you chose 3 and not more (reference needed to back up your decision)? What was the level of agreement among raters-experts? Report

You stated: ''The setting of cameras allowed continuous observation of the athletes,'' I would be more precise -try to rewrite in this manner - The setting of cameras around the fighting area (tatami) ... Also model of all equipment used should be reported - a model of cameras.

You stated: ''A single sheet was developed as the essential research tool''. So what was on that sheet? If it is essential, then report it (perhaps as an appendix or supplemental material), as I don't see how this study can be replicated without it!

The formulas section is still very unclear and hard to read. Restructure for greater clarity! At least add some empty lines before each new formula.

Your answer about body height and age should be in the paper and not just as an answer to the reviewer. ADD

The sentence ''The results are presented and compared in the tables 3-5'' does not fit into the methods! Delete

Line 196 - Firstly, it was verified whether both variables were normally distributed - you already reported that - Delete.

Lines 199-200 - Correlation analysis was applied in order to check the relationship between two continuous variables. Delete. It is well known why we use correlation statistics.

Table 3 - If I read your table correctly, there are several mistakes. The indicator is the same in all variables - Sa. Total activeness should be A, Total efficiency - Sk and so on. Correct

Also, add abbreviations for Q1 ME Q2 under the table (And in English, please)

Line 227-228: BD deleted in pairs, Coefficients in red are significant with p < 0.05  / What is BD abbreviation? In red? There is no colour - did you mean in bold? Correct and report

Line 275-276 -  the course of the sports training process. Really? How can you state that? How did you measure that? Your results do not support this conclusion. Delete

Line 367 - and strength and conditioning? And how does your research support this? Have you measured strength and conditioning in any way? NO - delete

Why is reference number 10 written in Polish? Correct and add an English title.

Overall major revisions are still needed for improvement and greater clarity of the paper.

Kind regards

Author Response

Dear Reviewer,

Thank you very much for your time and valuable comments, which all have been considered and incorporated. The detailed list of responses is given below. We hope that the modifications and explanation will be acceptable for you.

Yours sincerely,

Rydzik, corresponding author

Thank you for addressing the majority of the comments. However, some parts still need to be adequately addressed.

The abstract is poorly written. Rewrite and add the all-important findings of your research including that there were no statistically significant differences in the level of technical and tactical preparation despite the change of fighting rules that took place between the competitions.

A: This part has been corrected

Line 23-25: The respondents were at the highest world elite sports skill level and had had medal achievements at the World Championships and Olympic Games. / The sentence is too long and it does not clearly state what sample was used. Rewrite

A: This part has been corrected

Table 1 - Q1, ME, and Q2 abbreviations are in Polish? Translate

A: This part has been corrected
Table 2 - The head of the table is incorrect as you just state World Championships Rio. Correct and also add OG

A: This part has been corrected

You stated: ''Sports fight analysis was performed by three champion-level judo coaches'' So what were your criteria to be a champion-level judo coach? Why did you chose 3 and not more (reference needed to back up your decision)? What was the level of agreement among raters-experts? Report

A: There are four instructor and coach levels in judo in Poland. The lowest level is instructor, followed by the Level 2 judo coach, Level 1 judo coach, and champion-level coach (time of obtaining champion-level coach's degree from an instructor level is 10 years). To become a champion-level coach it is necessary to have the successes of athletes on the level of national championships, undergo a 60-hour course, pass an exam, and defend a thesis concerning Judo. The license of a champion-level coach is issued by the Ministry of Sport or Polish sports associations. Due to the prestige of having a champion-level coach (only several people in Poland), we considered three opinions sufficient.  The level of agreement was 95%.

You stated: ''The setting of cameras allowed continuous observation of the athletes,'' I would be more precise -try to rewrite in this manner - The setting of cameras around the fighting area (tatami) ... Also model of all equipment used should be reported - a model of cameras.

A: This part has been corrected as suggested.

You stated: ''A single sheet was developed as the essential research tool''. So what was on that sheet? If it is essential, then report it (perhaps as an appendix or supplemental material), as I don't see how this study can be replicated without it!

A: In our opinion, attaching the sheet is not necessary, but we have added a description. The sheet was individual support in developing a pattern of the fight.

The formulas section is still very unclear and hard to read. Restructure for greater clarity! At least add some empty lines before each new formula.

A: For greater clarity and easier reading, we have added empty lines between the formulas.

Your answer about body height and age should be in the paper and not just as an answer to the reviewer. ADD

A: Relevant information has been added.

The sentence ''The results are presented and compared in the tables 3-5'' does not fit into the methods! Delete

A: This sentence has been deleted.

Line 196 - Firstly, it was verified whether both variables were normally distributed - you already reported that - Delete.

Lines 199-200 - Correlation analysis was applied in order to check the relationship between two continuous variables. Delete. It is well known why we use correlation statistics.

A: This part has been removed

Table 3 - If I read your table correctly, there are several mistakes. The indicator is the same in all variables - Sa. Total activeness should be A, Total efficiency - Sk and so on. Correct

A: This part has been corrected

Also, add abbreviations for Q1 ME Q2 under the table (And in English, please)

A: the abbreviations have been added.

Line 227-228: BD deleted in pairs, Coefficients in red are significant with p < 0.05  / What is BD abbreviation? In red? There is no colour - did you mean in bold? Correct and report

A: This mistake has been corrected, we meant in bold

Line 275-276 -  the course of the sports training process. Really? How can you state that? How did you measure that? Your results do not support this conclusion. Delete

A: The sentence has been changed

Line 367 - and strength and conditioning? And how does your research support this? Have you measured strength and conditioning in any way? NO – delete

A: This part has been corrected

Why is reference number 10 written in Polish? Correct and add an English title.

A: This is the original title, we have added "in Polish"